

# Mislabeled and ambiguous market names in invertebrate and finfish seafood conceal species of conservation concern in Calgary, Alberta, Canada

Matthew R. J. Morris[1], Mindi M. Summers[2], Morgan Kwan[1], Jonathan A. Mee[3] and Sean M. Rogers[2]

[1] Department of Biology, Ambrose University, Calgary, Alberta, Canada
[2] Department of Biological Sciences, University of Calgary, Calgary, Alberta, Canada
[3] Department of Biology, Mount Royal University, Calgary, Alberta, Canada

Corresponding author
Matthew R. J. Morris,
Matthew.Morris@ambrose.edu

## ABSTRACT

**Background:** The mislabeling of seafood, wherein a food product's marketed name does not match its contents, has the potential to mask species of conservation concern. Less discussed is the role of legally ambiguous market names, wherein a single name could be used to sell multiple species. Here we report the first study in Canada to examine mislabeling and ambiguous market names in both invertebrate (*e.g.*, bivalve, cephalopod, shrimp) and finfish products.

**Methods:** A total of 109 invertebrate and 347 finfish products were sampled in Calgary between 2014 and 2020. Market names were documented from the label or equivalent and determined to be precise (the name could apply to only one species) or ambiguous (multiple species could be sold under that name). A region of the *cytochrome c oxidase I* gene was sequenced and compared to reference sequences from boldsystems.org. Samples were considered mislabeled if the species identified through DNA barcoding did not correspond to the market name, as determined through the Canadian Food Inspection Agency Fish List. Mislabeling was further differentiated between semantic mislabeling, wherein the market name was not found on the Fish List but the barcode identity was in line with what a consumer could reasonably have expected to have purchased; invalid market names, wherein the market name was so unusual that no legitimate inferences as to the product's identity could be made; and product substitution, wherein the DNA barcode identified the product as a species distinct from that associated with the market name. Invalid market names and product substitutions were used to provide conservative estimates of mislabeling. The global conservation status of the DNA-identified invertebrate or finfish was determined through the International Union for the Conservation of Nature Red List. A logistic regression was used to determine the relationship between precision and accuracy in predicting conservation status of the sampled species.

**Results:** There was no significant difference in mislabeling occurrence between invertebrates (33.9% total mislabeling occurrence, 20.2% product substitution) and finfish (32.3% total mislabeling occurrence, 21.3% product substitution/invalid market names). Product substitutions sometimes involved species of conservation concern, such as foods marketed as freshwater eel (*Anguilla rostrata*) that were

determined through DNA barcoding to be European eel (*Anguilla anguilla*), or cuttlefish balls putatively identified as the Endangered threadfin porgy (*Evynnis cardinalis*). Product substitutions and ambiguous market names were significantly associated with the sale of species of conservation concern, but ambiguity was a more important predictor. Although preventing the mislabeling of seafoods can and must remain a priority in Canada, our work suggests that moving towards precise names for all seafood products will better support sustainable fisheries goals.

## INTRODUCTION

Seafood, including marine and freshwater fishes and invertebrates, are one of the few wild sources of protein commercially harvested and globally distributed (*Ritchie & Roser, 2021*). Mismanagement of these resources, coupled with environmental stressors, have led to the collapse or imminent collapse of many wild seafood stocks (*Britten, Duarte & Worm, 2021*). Conservation groups interested in the fate of wild fishes have tried to help consumers make informed choices about which fish on the market have been sustainably harvested (*Winson et al., 2022*); however, the efficacy of this approach depends upon the reliability of the product names in the first place (*e.g., Barendse et al., 2019; Willette et al., 2021*). Seafood mislabeling, wherein the market name for the product does not match its contents, is a global phenomenon (*e.g., Hanner et al., 2011; Galal-Khallaf et al., 2014; Nagalakshmi et al., 2016; Carvalho et al., 2017; Nedunoori, Turanov & Kartavtsev, 2017; Chen et al., 2019; Do et al., 2019; Luque & Donlan, 2019; Kroetz et al., 2020; Minoudi et al., 2020; Wallstrom et al., 2020; Khalil, Gainsford & van Herwerden, 2023*) that may hamper conservation efforts.

The economic, ecological, cultural, and health effects of mislabeling are diverse and context-specific, but are typically negative (*e.g., Donlan & Luque, 2019; Williams, Hernandez-Jover & Shamsi, 2020; Morris, 2020; Silva, Hellberg & Hanner, 2021*). Investigations of mislabeling often detect at least a few species of conservation concern sold under market names that hide their identity (*Marchetti et al., 2020; Silva, Hellberg & Hanner, 2021; Nijman & Stein, 2022; Khalil, Gainsford & van Herwerden, 2023*), causing informed consumers to purchase endangered species despite their best intentions. Conversely, farmed products may be disguised as wild-caught species, hiding from the consumer the economic and/or environmental damages associated with farming (*Korzik et al., 2020*) or giving the impression that the wild species being marketed is not in need of conservation (*Cawthorn, Baillie & Mariani, 2018*). The extent to which conservation concerns are comparable between invertebrate and finfish mislabeling is largely unknown given the paucity of data on invertebrates (*Luque & Donlan, 2019*).

In Canada, seafood product labeling falls under the authority of the Canadian Food Inspection Agency (CFIA), whose Fish List (https://active.inspection.gc.ca/scripts/fssa/fispoi/fplist/fplist.asp?lang=e) guides vendors towards "acceptable" market names to

prevent labeling that is "false, misleading or deceptive" (*Canadian Food Inspection Agency (CFIA), 2019a*, *2019b*, CFIA's words in quotations). The CFIA Fish List leaves room for ambiguity in market names, wherein a single market name could acceptably be used for more than one species. Examples include the market names of snapper, tuna, and cod, which could acceptably be used in Canada for the sale of 96, 14, and two species, respectively. Ambiguity may be an additional important factor in mislabeling and conservation. Ambiguity should reduce mislabeling, as it gives room for vendors to sell species whose exact identity is unknown, while it may harm conservation efforts by permitting consumers to unknowingly purchase species of conservation concern (*Cawthorn, Baillie & Mariani, 2018*; *Cawthorn et al., 2021*; *Hu et al., 2018*).

Mislabeled seafood has been detected in coastal and continental regions of Canada (*e.g.*, *Wong & Hanner, 2008*; *Hanner et al., 2011*; *Hu et al., 2018*; *Levin, 2018*; *Shehata et al., 2018*; *Cawthorn et al., 2021*), but an extensive examination of mislabeling in the prairie provinces has not been done (but see *Morris, 2020* for an initial exploration of some of the finfish data used in this article, in the context of addressing the concerns of a faith-based audience surrounding mislabeling). In this article we summarize the work done at three different academic institutions in Calgary, Alberta, Canada, to document and compare mislabeling in a prairie city in a landlocked province. Here we investigate (1) the extent to which mislabeling occurs in Calgary, and how it compares to other Canadian cities; (2) how mislabeling compares between marine invertebrates and finfish; (3) whether mislabeling hides species of conservation concern; and (4) the role that ambiguous market names may play in facilitating mislabeling or masking species of conservation concern.

## MATERIALS AND METHODS

Between 2014 and 2020, 345 students (graduate and undergraduate) at three separate institutions in Calgary, Alberta, Canada (Ambrose University, Mount Royal University, and University of Calgary) sampled seafood as part of a class assignment for genetics, molecular genetics, invertebrate zoology, or independent research courses. Samples were selected from food vendors in and around Calgary. Students collected product data following *Naaum et al. (2015)*, including the market name under which it was sold, the type of sample (*e.g.*, whole, fillet, canned, dried), its processing level (*e.g.*, head off, frozen status), the vendor type (*e.g.*, grocery store, restaurant, fish market), the vendor name and address, the price (in $ per unit weight if known), and additional details such as whether the product was labeled as farmed or wild (File S1—please note some details, such as address, have been withheld). Five additional samples were collected in Calgary in 2017 by SeaChoice (*Lifescanner & SeaChoice, 2017*). Students sampled seafood from food vendors in September or early October at the start of fall semester, or in January or early February at the start of winter semester. Students were not permitted to sample the same fish product from the same vendor in the same sampling year. Students were given single-use tweezers and alcohol wipes with instructions to sterilize their tweezers on location, and use the tweezers to place a single piece of tissue, no smaller than a kidney bean, into a Lifescanner vial. In most cases, product including the market name was photographed before being placed in the vial, and photographed again in the vial. Whenever possible,

photographs were uploaded to the Barcode of Life Data Systems (BOLD) website in association with the DNA sequence. Vials were shipped to the University of Guelph to be sequenced through the Biodiversity Institute of Ontario or, after Lifescanner underwent a change of ownership, Biolytica Inc. *COI* sequences were identified to species using the boldsystems.org identification algorithm, where they were compared to sequences available in the user-supplied database.

A total of 85% of finfish and 72% of invertebrate samples provided DNA sequences (excluding Mount Royal samples, for which failed sequence data was not available); others were too short to be usable, or did not amplify. Canned fish products in particular rarely returned usable DNA. Of the sequences that were returned, five (three invertebrates, two finfish) could not be resolved to species even after trimming the sequence. Three additional sequences were suggestive of contamination (*e.g.*, a sample of prawn was barcoded as Atlantic salmon but only from a partial barcode of 122 nucleotides in length). The remaining 347 finfish and 109 invertebrate seafood products returned DNA barcodes that could be analyzed. Collectively, sampling effort varied by year (Table 1). One student-led independent research project focused on invertebrates in the spring of 2017, and included replicates (sampling the same tissue, or multiple individuals from the same package); replicates are not included in total counts.

Mislabeling was defined according to the CFIA Fish List. Each market name (the name on the product label, menu, or equivalent) was recorded and compared to the Fish List to determine which species could acceptably be sold under that name. These acceptable species names were checked against the DNA barcode identity of the product. A product was determined to be properly labeled if: (1) the market name and DNA barcode identity were the same as determined through the Fish List; or (2) the market name corresponded with at least one of the several species that matched the DNA barcode. This second case could include genuine forms of mislabeling, but the DNA barcode could not give the resolution to detect it. Mislabeling was categorized as follows: (1) Semantic mislabeling, wherein the market name was not found on the Fish List but the barcode identity was in line with what a consumer could reasonably have expected to have purchased. For example, a consumer could reasonably expect to be consuming American eel, *Anguilla rostrata*, when purchasing "unagi" or "freshwater eel", despite neither market name being found on the Fish List (the Fish List recognizes "eel" or "American eel" as acceptable market names). (2) Invalid market names, wherein the market name was so unusual that no legitimate inferences as to the product's identity could be made—meaning no appeal to the Fish List could be used. (3) Product substitution, wherein the DNA barcode identified the product as a species distinct from that associated with the market name. Invalid market names and product substitutions were used to provide conservative estimates of mislabeling.

Samples were grouped into broad taxonomic categories (family level for finfish, class for invertebrates), based on their market names. Due to shared labels between rockfishes and snappers, Lutjanidae and Scorpaenidae were grouped together. Finfish classification was determined according to Eschmeyer's Catalog of Fishes (*Fricke, Eschmeyer & Van der Laan, 2024*).

**Table 1 Mislabeling of invertebrate and finfish products by year.** S, semantic mislabeling (wherein the market name was not found on the Fish List but the barcode identity was in line with what a consumer could reasonably have expected to have purchased). IN, invalid market name (wherein the market name was so unusual that no legitimate inferences as to the product's identity could be made). PS, product substitution (wherein the DNA barcode identified the product as a species distinct from that associated with the market name). % Mis (All), percentage of samples that were mislabeled. % Mis (Cons), conservative estimate of mislabeling, including only invalid market names and product substitutions.

| | Invertebrates | | | | | | Finfish | | | | | |
|---|---|---|---|---|---|---|---|---|---|---|---|---|
| Year | No. sampled | S | IN | PS | % Mis (All) | % Mis (Cons) | No. sampled | S | IN | PS | % Mis (All) | % Mis (Cons) |
| 2014 | 0 | 0 | 0 | 0 | NA | NA | 19 | 0 | 0 | 6 | 31.6 | 31.6 |
| 2016 | 1 | 0 | 0 | 1 | 100 | 100 | 57 | 5 | 0 | 12 | 29.8 | 21.1 |
| 2017 | 1 | 0 | 0 | 1 | 100 | 100 | 132 | 19 | 6 | 28 | 40.2 | 25.8 |
| 2018 | 47 | 4 | 0 | 14 | 38.3 | 29.8 | 62 | 10 | 0 | 10 | 32.2 | 16.1 |
| 2019 | 16 | 3 | 0 | 3 | 37.5 | 18.8 | 60 | 4 | 2 | 7 | 21.7 | 15.0 |
| 2020 | 44 | 8 | 0 | 3 | 25.0 | 6.8 | 17 | 0 | 0 | 3 | 17.6 | 17.6 |
| Total | 109 | 15 | 0 | 22 | 33.9 | 20.2 | 347 | 38 | 8 | 66 | 32.3 | 21.3 |

Market names were categorized as ambiguous if the Fish List recommended the sale of multiple species under that particular market name; conversely, market names were categorized as precise if no more than one species could be sold under that market name. In cases where an entire genus could be sold under a particular market name, that name was considered ambiguous. Ambiguous names also included market names not found on the CFIA Fish List that nonetheless could refer to multiple species (*e.g.*, "Pacific rockfish", "red tuna").

Each product was assigned an International Union for the Conservation of Nature (IUCN) global conservation status using the Red List (www.iucnredlist.org/) based on its barcode identity, including Data Deficient, Least Concern, Near Threatened, Vulnerable, Endangered, or Critically Endangered, or Not Applicable if the species could not be found on the IUCN Red List. If the DNA barcode was a match to multiple species when compared against the Barcode of Life Data System (BOLD) database, the species with the highest data-based conservation status was used. That is, if four species identities were associated with a single DNA barcode, and they were ranked as Not Applicable, Data Deficient, Least Concern, and Critically Endangered, Critically Endangered was used for statistical tests.

Pearson's Chi-square tests were performed in R v.4.0.3 (*R Core Team, 2020*) to determine if invertebrates and finfish had similar levels of both mislabeling and ambiguous market names. Chi-square or Fisher's exact tests were conducted, depending on sample size, to test the null hypothesis that there was a random association between the following: conservation status (least concern *versus* conservation concern = vulnerable or higher) and mislabeling presence/absence; ambiguity in market names and mislabeling presence/ absence; and ambiguity in market names and conservation status, for invertebrates and finfish respectively. To determine the combined effect of precision (ambiguity) and accuracy (mislabeling) on conservation status, data for finfish was converted into dichotomous nominal data for conservation status (0 = least concern, 1 = conservation

concern), precision (0 = precisely labeled, 1 = ambiguously labeled), and accuracy (0 = correctly labeled, 1 = mislabeled). Data was fit to a logistic regression using precision, accuracy, and their interactions to predict conservation status. Models with and without the interaction term were first compared using Akaike Information Criteria generated in the *AICcmodavg* package (v.2.3.1; *Mazerolle, 2023*). Multicollinearity was tested using the *car* package (v.3.0.10; *Fox & Weisberg, 2019*), to ensure that variables had variable inflation factors less than five. McFadden's pseudo-$R^2$ was calculated using the *pscl* package (v.1.5.5; *Jackman, 2020*) and variable importance was determined using the *caret* package (v.6.0.86; *Kuhn, 2008*).

## RESULTS

### Market names

Students sampled products that were sold under 33 market names for invertebrates and 78 market names for finfish (Tables 2, 3). Amongst invertebrates, the most common market names were shrimp ($n = 20$), followed by octopus ($n = 12$), Pacific white shrimp ($n = 10$), and squid ($n = 9$). Amongst finfish, the most common market names were salmon ($n = 39$), followed by Atlantic salmon ($n = 30$), tuna ($n = 28$), and sockeye salmon ($n = 20$).

Two samples each of finfish and invertebrates were sold under market names that were not regulated by the Fish List (*e.g.*, mixed products labeled "seafood medley" are legally permitted but fall under a different set of guidelines). Barring those, 15.0% ($n = 16$ samples, sold under 11 market names) of invertebrate and 13.0% ($n = 45$ samples, sold under 24 market names) of finfish were sold using names not found on the Fish List, not including misspelled but otherwise acceptable names (invertebrates: $n = 4$ misspelled products; finfish: $n = 25$ misspelled products) (Tables 2, 3). Most market names were provided in print form (*e.g.*, menus, sushi order sheets, sticker labels, receipts), with two invertebrate and nine finfish names provided orally or online (*e.g.*, Filet-O-Fish identified online as Alaskan pollock).

Invertebrate products showed particularly high levels of ambiguous market names (81.6%, $n = 89$) compared to finfish (50.7%, $n = 176$) ($X^2 = 33.5$, df = 1, $p = 7.06 \times 10^{-9}$).

### DNA barcoding results

DNA barcodes were on average 554 nucleotides long for invertebrates (range: 100–651 nucleotides), and 559 nucleotides long for finfish (range: 99–700 nucleotides). Of the 109 invertebrate and 347 finfish that were successfully barcoded, 83 invertebrate and 186 finfish samples identified to a single species. The remainder, 26 invertebrates and 161 finfish, matched to more than one species on the BOLD database.

### Mislabeling: overview

Occurrences of mislabeling were similar among invertebrates and finfish, with a total mislabeling rate of 33.9% for invertebrates (Tables 1, 2, Fig. 1) and 32.3% for finfish ($X^2 = 0.04$, df = 1, $p$-value = 0.8) (Tables 1–3, Fig. 1). Mislabeling was conservatively estimated at 20.2% product substitution for invertebrates (there were no instances of invalid names) and 21.3% invalid names/product substitution for finfish (19.0% product

**Table 2 Mislabeling by market name in invertebrates.** The number of mislabeled products, and the Canadian Food Inspection Agency approved name(s) for the DNA-barcoded specimen, are provided. If the species could not legally be sold in Canada, the Latin name of the species is provided; if DNA barcodes returned more than one species, only the CFIA-approved species are included. Semantic mislabeling was defined as occurring when the market name was not found on the Fish List but the barcode identity was in line with what a consumer could reasonably have expected to have purchased. Invalid market name was defined as occurring when the market name was so unusual that no legitimate inferences as to the product's identity could be made. Product substitution was defined as occurring when the DNA barcode identified the product as a species distinct from that associated with the market name.

| Market name | Sample size | Semantic mislabeling | Invalid market name | Product substitution |
|---|---|---|---|---|
| Argentine Prawn | 1 | 1–Shrimp | 0 | 0 |
| Atlantic Mussel | 2 | 2–Mussel or Atlantic Bay Mussel | 0 | 0 |
| Atlantic Scallop | 1 | 0 | 0 | 0 |
| Black Tiger Shrimp | 2 | 0 | 0 | 0 |
| Calamari | 1 | 0 | 0 | 0 |
| California Squid | 1 | 1–California Market Squid | 0 | 0 |
| Cherrystone Clam | 1 | 1–Cherrystone or Clam | 0 | 0 |
| Clam | 2 | 0 | 0 | 0 |
| Colossal Freshwater Shrimp | 1 | 1–Freshwater Shrimp | 0 | 0 |
| Crab | 6 | 0 | 0 | 1–Threadfin Bream; 1–Bigeye; 1–Red Crab |
| Cuttlefish | 4 | 0 | 0 | 1–Bobtail Squid; 1–Yellowback Seabream; 1–Bocourti Catfish |
| Giant Squid | 1 | 1–Squid or Calamari or Japanese Flying Squid | 0 | 0 |
| King Crab | 1 | 0 | 0 | 1–Pacific Hake or relative |
| Lobster | 3 | 0 | 0 | 1–Spiny Lobster or Crayfish |
| Malpeque Oyster | 4 | 3–Atlantic Oyster | 0 | 1–Pacific Oyster |
| Malpeques | 1 | 1–Atlantic oyster | 0 | 0 |
| Manila Clam | 1 | 0 | 0 | 0 |
| Mussel | 6 | 0 | 0 | 0 |
| Octopus | 12 | 0 | 0 | 1–*Amphioctopus* sp.; 1–*Amphioctopus aegina*; 1–Squid or Jumbo Squid or Calamari; 2–Calamari or Japanese Flying Squid or Squid |

(Continued)

| Market name | Sample size | Semantic mislabeling | Invalid market name | Product substitution |
|---|---|---|---|---|
| Oyster | 1 | 0 | 0 | 0 |
| Pacific White Shrimp | 10 | 0 | 0 | 0 |
| Prawn | 3 | 0 | 0 | 0 |
| Scallop | 2 | 0 | 0 | 0 |
| Seafood Medley/Mix | 2 | 0 | 0 | 0 |
| Shrimp | 20 | 0 | 0 | 1–*Solenocera crassicornis* |
| Snow Crab | 1 | 0 | 0 | 1–Red Crab |
| Southern King Crab | 1 | 0 | 0 | 0 |
| Squid | 9 | 0 | 0 | 1–*Sepia aculeata*; 1–*Sepia recurvirostra* |
| Squid/Cuttlefish | 1 | 0 | 0 | 0 |
| Surf Clam | 4 | 0 | 0 | 4–*Pseudocardium sachalinense* |
| Tako | 2 | 2–Octopus or Common Octopus | 0 | 0 |
| Torpedo Bay Oyster | 1 | 1–Oyster or Pacific Oyster | 0 | 0 |
| Uni | 1 | 1–(if *Strongylocentrotus purpuratus*): Sea Urchin or Purple Sea Urchin; (if *Mesocentrotus franciscanus*): Sea Urchin or Red Sea Urchin | 0 | 0 |

**Table 3 Mislabeling by market name in finfish.** The number of mislabeled products, and the Canadian Food Inspection Agency approved name(s) for the DNA-barcoded specimen, are provided. If the species could not legally be sold in Canada, the Latin name of the species is provided; if DNA barcodes returned more than one species, only the CFIA-approved species are included. Semantic mislabeling was defined as occurring when the market name was not found on the Fish List but the barcode identity was in line with what a consumer could reasonably have expected to have purchased. Invalid market name was defined as occurring when the market name was so unusual that no legitimate inferences as to the product's identity could be made. Product substitution was defined as occurring when the DNA barcode identified the product as a species distinct from that associated with the market name.

| Market name | Sample size | Semantic mislabeling | Invalid market name | Product substitution |
|---|---|---|---|---|
| Ahi Tuna | 6 | 3–Tuna or Yellowfin or Yellowfin Tuna | 0 | 1–(if *Thunnus thynnus*): Tuna or Atlantic Bluefin Tuna or Northern Bluefin Tuna or Bluefin Tuna; (if *Thunnus orientalis*): Tuna, Pacific Bluefin Tuna, Bluefin Tuna, Oriental Tuna; 2–Tuna or Bigeye Tuna |
| Alaska Pollock | 5 | 0 | 0 | 0 |

| Market name | Sample size | Semantic mislabeling | Invalid market name | Product substitution |
|---|---|---|---|---|
| Alaska Pollock/ Pacific Whiting (Imitation Crab) | 1 | 0 | 0 | 0 |
| Alaskan Cod | 1 | 0 | 1–(if *Boreogadus saida*): Arctic Cod or Polar Cod; (if *Gadus ogac*): Greenland Cod or Ogac; (if *G. macrocephalus*): Grey Cod or Cod or Pacific Cod | 0 |
| Alaskan Salmon | 2 | 0 | 1–Sockeye Salmon or Red Sockeye Salmon or Red Salmon | 1–Rockfish |
| Albacore Tuna | 2 | 0 | 0 | 1–Tilapia |
| Amberjack | 1 | 0 | 0 | 0 |
| Arctic Char | 1 | 0 | 0 | 0 |
| Atlantic Cod | 1 | 0 | 0 | 1–Grey Cod or Cod or Pacific Cod |
| Atlantic Salmon | 30 | 0 | 0 | 0 |
| Basa | 15 | 0 | 0 | 0 |
| BF Negi Toro | 1 | 1–Tuna or Bluefin Tuna | 0 | 0 |
| Bigeye Tuna | 1 | 0 | 0 | 0 |
| Bluefin Tuna | 2 | | 0 | 0 |
| Butterfish/ Oilfish | 1 | 0 | 0 | 1–Snake Mackerel or Escolar |
| Canadian Lake Whitefish | 1 | 1–Lake Whitefish or Whitefish | 0 | 0 |
| Chinook Salmon | 1 | 0 | 0 | 0 |
| Cod | 19 | 0 | 0 | 1–Arctic Charr or Arctic Char or Char; 1–Blue Whiting or Southern Blue Whiting or Blue Cod |
| Coho Salmon | 1 | 0 | 0 | 0 |
| Conger Eel | 1 | 0 | 0 | 0 |
| Corvina | 1 | 0 | 0 | 1–Croaker or Whitemouth Drummer |
| Dancing Eel | 1 | 1–Eel or American Eel | 0 | 0 |
| Eel | 2 | 0 | 0 | 1–*Anguilla anguilla* |
| Freshwater Eel | 5 | 4–Eel or American Eel | 0 | 1–*Anguilla anguilla* |
| Freshwater Smelt | 1 | 1–American Smelt or Lake Smelt or Rainbow Smelt or Smelt | 0 | 0 |
| Golden Threadfin Bream | 1 | 0 | 0 | 1–Japanese Threadfin Bream |

(Continued)

| Market name | Sample size | Semantic mislabeling | Invalid market name | Product substitution |
|---|---|---|---|---|
| Haddock | 3 | 0 | 0 | 0 |
| Halibut | 13 | 0 | 0 | 0 |
| Hamachi | 1 | 1–Japanese Amberjack | 0 | 0 |
| Hamachi–Yellow Tail Tuna | 1 | 0 | 1–Japanese Amberjack | 0 |
| Hamachi–Yellowtail Jackfish | 1 | 0 | 1–Japanese Amberjack | 0 |
| Imitation Crab | 2 | 0 | 0 | 0 |
| Jack Mackerel | 1 | 0 | 0 | 0 |
| Japanese Mackerel | 1 | 0 | 1–Mackerel or Atlantic Mackerel | 0 |
| Mackerel | 10 | 0 | 0 | 0 |
| Mahi-mahi | 1 | 0 | 0 | 0 |
| Marlin | 1 | 0 | 0 | 1–Blue Marlin |
| New Zealand Blue Cod | 2 | 2–Blue Whiting or Southern Blue Whiting or Blue Cod | 0 | 0 |
| North Atlantic Haddock | 1 | 1–Haddock | 0 | 0 |
| Ocean Perch | 1 | 0 | 0 | 0 |
| Pacific Cod | 11 | 0 | 0 | 3–Cod or Atlantic Cod |
| Pacific Halibut | 1 | 0 | 0 | 0 |
| Pacific Herring | 1 | 0 | 0 | 0 |
| Pacific Rockfish | 2 | 1–(if *Sebastes flavidus*): Pacific Snapper or Yellowtail Rockfish or Rockfish; (if *S. serranoides*): Olive Rockfish or Rockfish; 1–(if *Sebastes pinniger*): Pacific Snapper or Canary Rockfish or Rockfish; (if *Sebastes* sp.): Rockfish | 0 | 0 |
| Pacific Salmon | 1 | 0 | 1–Pacific Pink Salmon or Pink Salmon | 0 |
| Pacific Snapper | 7 | 0 | 0 | 1–Rockfish |
| Pacific Sole | 1 | 1–(if *Limanda aspera*): Sole or Yellowfin Sole or Flounder; (if *L. limanda*): Common Dab | 0 | 0 |

| Market name | Sample size | Semantic mislabeling | Invalid market name | Product substitution |
|---|---|---|---|---|
| Pacific Yellowtail Snapper | 2 | 0 | 0 | 1–(if *Sebastes brevispinis*): Rockfish or Silvergray Rockfish or Pacific Snapper; (if *S. zacentrus*): Rockfish or Sharpchin Rockfish; (if *S. borealis*): Rockfish or Rosefish or Pacific Snapper or Shortraker Rockfish; 1–(if *Sebastes flavidus*): Pacific Snapper or Yellowtail Rockfish or Rockfish; (if *S. serranoides*): Olive Rockfish or Rockfish |
| Pickerel | 4 | 0 | 0 | 0 |
| Pink Salmon | 2 | 0 | 0 | 0 |
| Pollock | 2 | 0 | 0 | 1–Sole or Yellowfin Sole or Flounder |
| Ponyfish | 1 | 0 | 0 | 0 |
| Red Snapper | 16 | 0 | 0 | 1–Snapper or Malabar Snapper or Malabar Blood Snapper; 1–(if *Lutjanus russellii*): Snapper or Russell's Snapper; (if *L. johnii*): Snapper or John's Snapper; 1–(if *Sebastes flavidus*): Pacific Snapper or Yellowtail Rockfish or Rockfish; (if *S. serranoides*): Olive Rockfish or Rockfish 13–Tilapia |
| Red Tuna | 8 | 8–Tuna | 0 | 0 |
| Salmon | 39 | 0 | 0 | 1–(if *Thunnus alalunga*): Tuna or Albacore Tuna; (if *T. obesus*): Tuna or Bigeye Tuna; 2–White Chinook or Chinook Salmon or King Salmon or Pink Chinook or Red Chinook or Spring Salmon or Chinook; 2–Pacific Pink Salmon or Pink Salmon; 9–Steelhead Trout or Deep Sea Trout or Rainbow Trout or Steelhead Salmon or Trout or Steelhead; 1–Sockeye Salmon or Red Sockeye Salmon or Red Salmon |
| Saltwater Eel | 1 | 0 | 1–Eel or American Eel | 0 |
| Sea Bass | 1 | 0 | 0 | 1–Chum Salmon or Keta Salmon or Silverbrite Salmon |
| Sea Eel | 1 | 0 | 1–*Ophichthus remiger* | 0 |
| Short Mackerel | 1 | 0 | 0 | 0 |
| Skipjack Tuna | 1 | 0 | 0 | 0 |
| Snapper | 8 | 0 | 0 | 2–(if *Sebastes flavidus*): Pacific Snapper or Yellowtail Rockfish or Rockfish; (if *S. serranoides*): Olive Rockfish or Rockfish; (if *S. mystinus*): Blue Rockfish or Rockfish; 6–Tilapia |
| Sockeye Salmon | 20 | 0 | 0 | 0 |

*(Continued)*

| Market name | Sample size | Semantic mislabeling | Invalid market name | Product substitution |
|---|---|---|---|---|
| Sole | 6 | 0 | 0 | 0 |
| Steelhead Salmon | 3 | 0 | 0 | 0 |
| Steelhead Trout | 4 | 0 | 0 | 0 |
| Tilapia | 7 | 0 | 0 | 0 |
| Trout | 1 | 0 | 0 | 0 |
| Tuna | 28 | 0 | 0 | 1–Snake Mackerel or Escolar |
| Unagi | 1 | 1–Eel or Freshwater Eel | 0 | 0 |
| Walleye | 3 | 0 | 0 | 0 |
| White Bass | 1 | 0 | 0 | 0 |
| White Tuna | 2 | 2–Tuna | 0 | 0 |
| Whitefish | 1 | 0 | 0 | 1–Arctic Charr or Arctic Char or Char |
| Yellow Croaker | 1 | 0 | 0 | 0 |
| Yellowfin Tuna | 3 | 0 | 0 | 0 |
| Yellowtail | 10 | 9–Japanese Amberjack | 0 | 1–Tilapia |
| Yellowtail Rockfish | 1 | 0 | 0 | 1–Ocean Perch or Pacific Ocean Perch or Rosefish or Redfish or Rockfish |
| Yellowtail Snapper | 1 | 0 | 0 | 1–(if *Sebastes flavidus*): Pacific Snapper or Yellowtail Rockfish or Rockfish; (if *S. serranoides*): Olive Rockfish or Rockfish |

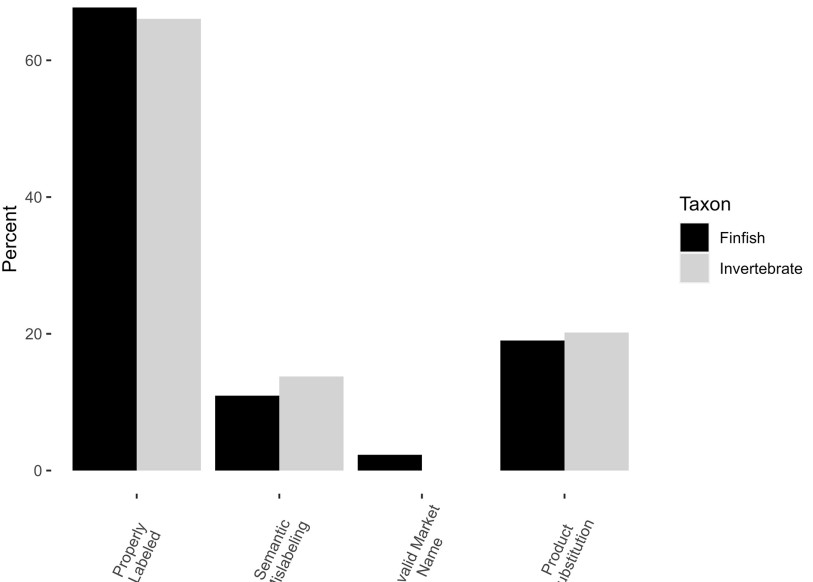

**Figure 1 Comparison of different forms of mislabeling.** Percentages of properly labeled and mislabeled finfish are shown in black, invertebrates in grey.
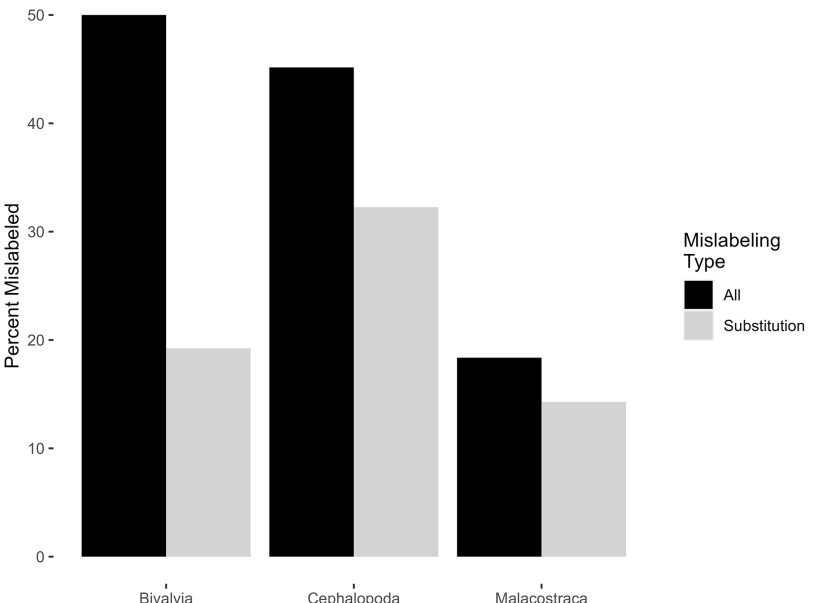

**Figure 2** **Percentage of mislabeled products amongst classes of invertebrates.** Percentage of mislabeled products is shown in black for all forms of mislabeling, and grey for product substitution only.

substitution alone), with no significant difference between finfish and invertebrates ($X^2$ = 0.01, df = 1, $p$ = 0.9). Ignoring semantics, mislabeling estimates dropped from approximately 1 in 3 to 1 in 5 for both invertebrates and finfish.

## Mislabeling: invertebrates

Invertebrate mislabeling varied by class (Fig. 2), with semantic mislabeling having a disproportionate effect on bivalves. 50% of bivalves (13 of 26), 45.2% of cephalopods (14 of 31), and 18.4% of malacostracans (nine of 49) fulfilled some definition of mislabeling; excluding semantic mislabeling, this dropped to product substitutions of 19.2%, 32.2%, and 14.3% respectively (Table 1). All four samples of surf clam were identified as *Pseudocardium sachalinense* (synonym of *Spisula sachalinense*)—a species not included in the Fish List. One Malpeque oyster (not a Fish List label, but which refers to Atlantic oyster) was actually Pacific oyster (*Magallana gigas*). Three of four cuttlefish samples were mislabeled, two being a species of finfish (one was bocourti—either *Pangasius bocourti*, *P. djambal*, or *Pangasianodon hypophthalmus*, and the other seabream/porgy—either *Dentex tumifrons, Parargyrops edita*, or *Evynnis cardinalis*), and one being *Sepiella inermis*, a cuttlefish not found on the Fish List. Five of 12 octopus samples were mislabeled, two being the squid *Todarodes pacificus*, one being a species of *Dosidicus*, and two belonging to the genus *Amphioctopus*, which is not found on the Fish List. Two of 13 squid were mislabeled, each belonging to a different species of cuttlefish (*Sepia aculeata* and *Sepia recurvirostra*). For malacostracans, one of 37 shrimp/prawn/tiger shrimp/freshwater shrimp samples was mislabeled, returning a DNA barcode of *Solenocera crassicornis* (a type of mud shrimp not found on the CFIA Fish List). Five of nine samples of crab/king crab/snow crab were mislabeled. Three of these were identified as species of finfish. One sample marketed as

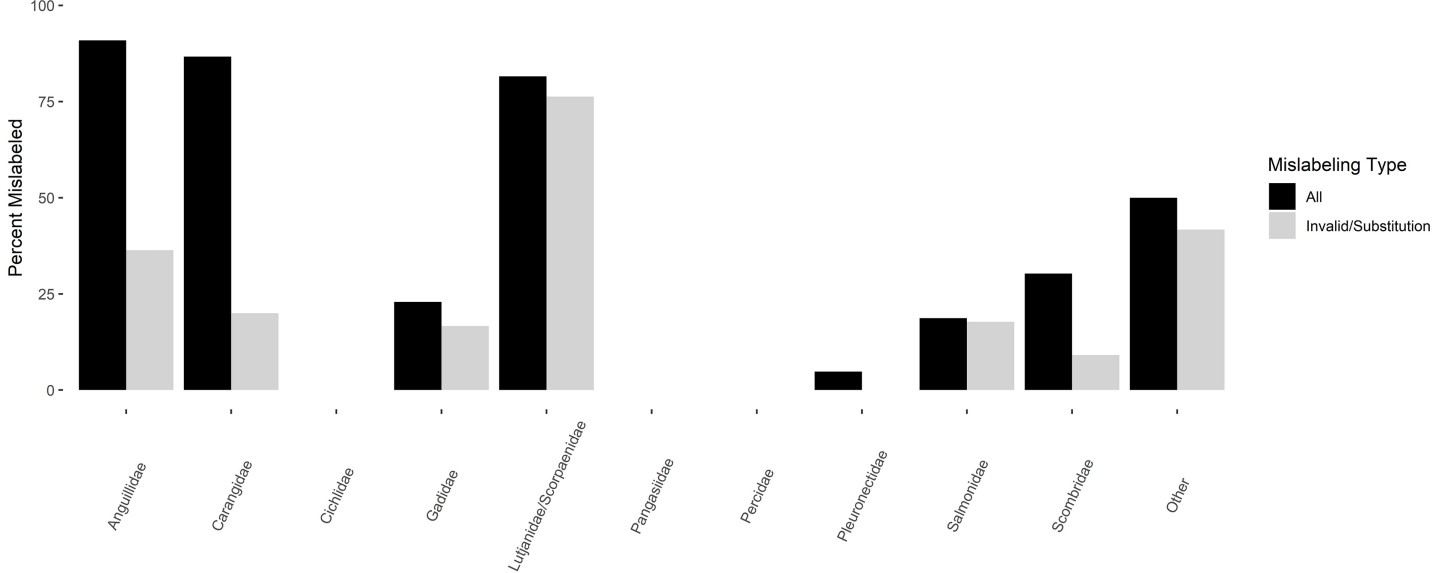

**Figure 3 Percentage of mislabeled products amongst families of finfishes.** Percentage of mislabeled products is shown in black for all forms of mislabeling, and grey for invalid market names/product substitution only.

crab had a DNA identity of *Nemipterus hexodon*, and another as *Priacanthus* (*P. hamrur* or *P. prolixus*). One sample of king crab was a species of hake (either *Merluccius angustimanus*, *M. productus*, or *M. gayi*). One crab and one snow crab sample returned DNA identities as either *Chaceon quinquedens* or *C. chuni*, only the first of which is on the Fish List and which is marketed as red crab or deepsea red crab. Finally, one of three lobsters was mislabeled, being *Panulirus argus*, which can be sold as crayfish or spiny lobster.

By far the most commonly encountered species was *Penaeus vannamei*, sold under the market names of shrimp, prawn, and Pacific white shrimp. Although the Fish List records 42 different species that can be marketed as shrimp, 19 species as prawn, and two species as Pacific white shrimp, 31 of 35 samples sold under those market names were identified as *Penaeus vannamei*. Not a single sample of *Penaeus vannamei* was mislabeled (Table 2).

Eleven invertebrate samples were divided into two replicates (either from the same tissue, or from the same package) to see how their sequences would compare. In each case, replicates agreed on their DNA barcoding identification. Three of these replicates confirmed cases of mislabeling, including one instance of crab that was actually a finfish.

## Mislabeling: finfish

A total of 96.5% of product names had Fish List identities corresponding to ten families of fish (Table 3, Fig. 3). Mislabeling varied by family (Fig. 3), with semantic mislabeling having a large impact on members of Scombridae, Carangidae, and Anguillidae, largely due to sushi names associated with these products that are not found on the Fish List (*e.g.*, "ahi tuna", "Hamachi", "unagi"). Product substitution and invalid market name mislabeling varied among families, being highest in Lutjanidae/Scorpaenidae (76.3%, 29 of 38 samples), followed by Anguillidae (36.4%, four of 11 samples), Carangidae (20%, three

of 15 samples), Salmonidae (17.8%, 19 of 107 samples), Gadidae (16.7%, eight of 48 samples), and Scombridae (9.1%, 6 of 66 samples). No product mislabeling was detected in Pleuronectidae (*n* = 21 samples), Pangasiidae (*n* = 15 samples), Cichlidae (*n* = 7 samples), or Percidae (*n* = 7 samples). The remaining 12 samples were spread among ten families, with an overall mislabeling of 41.7% (five of 12 samples).

Notable examples of mislabeling (Table 3) included products sold as sea eel barcoded as punctuated snake eel, *Ophichthus remiger*—a species not found on the Fish List for sale in Canada; pollock barcoded as yellowfin sole, *Limanda asper*; cod barcoded as southern blue whiting, *Micromesistius australis*; nine samples of salmon barcoded as rainbow trout, *Oncorhynchus mykiss*; one tuna, 19 snapper/red snapper, and one yellowtail all barcoded as tilapia, *Oreochromis* spp.; two samples of Pacific cod barcoded as Atlantic cod, *Gadus morhua*; one butterfish/oilfish and one tuna each barcoded as escolar, *Lepidocybium flavobrunneum*; and two eel barcoded as European eel, *Anguilla anguilla*. Ten of the 29 product substitutions within Lutjanidae/Scorpaenidae were not *Oreochromis* but instead were other members of the genus *Sebastes* or *Lutjanus*.

## Mislabeling and conservation

A total of 27 of the 109 invertebrate samples could, using barcode identities, be associated with at least one IUCN conservation status between Least Concern and Critically Endangered (*n* = 25 Least Concern, 2 Endangered). A total of 73 samples had barcode identities not found on the IUCN Red List, with the remainder being Data Deficient. A total of 276 of the 347 finfish samples could, using barcode identities, be associated with at least one IUCN conservation status between Least Concern and Critically Endangered (*n* = 142 Least Concern, 25 Near Threatened, 77 Vulnerable, 30 Endangered, and two Critically Endangered). There was not a significant association between product substitution presence/absence and conservation status in invertebrates (Fisher's exact test: *p*-value = 0.08, 95% CI for Odds Ratio: 0.5-infinity), although the only two species of conservation concern were mislabeled (both cuttlefish balls, each identified to multiple species of fish, including the Endangered *Pangasianodon hypophthalmus* for one and *Evynnis cardinalis* for the other). There was a significant association between conservative estimates of mislabeling and conservation concern in finfish ($X^2 = 8$, df = 1, *p* = 0.005) with mislabeled products being more likely to include species of conservation concern. Examples included the Critically Endangered European eel (*Anguilla anguilla*) sold as American eel (*Anguilla rostrata*), Vulnerable Atlantic cod (*Gadus morhua*) sold as Pacific cod (*Gadus microcephalus*), and the potential sale of the Vulnerable bigeye tuna (*Thunnus obesus*) as ahi tuna, which is traditionally yellowfin tuna (*Thunnus albacares*). Note we were not concerned here with whether a species of lesser conservation concern was substituted for one of greater conservation concern, but whether mislabeling had the potential to hide species of conservation concern in general. Ambiguous market names make looking for relative changes to conservation status challenging.

**Table 4 Results of a logistic regression of precision and accuracy on conservation status.** Precision (ambiguous or precise market names), accuracy (precisely labeled or mislabeled, using conservative estimates of mislabeling) and their interaction were used to predict conservation status (least concern or conservation concern).

| Coefficient | β | St. Error | Z-value | Pr (>|z|) | Odds ratio | 95% CI | Overall importance |
|---|---|---|---|---|---|---|---|
| Model 1: using all forms of ambiguity, with interaction term (AIC = 265, McFadden's pseudo-$R^2$ = 0.31, residual df = 267). | | | | | | | |
| Intercept | −1.99 | 0.29 | −6.97 | $3.2 \times 10^{-12}$ | | | |
| Precision | 3.22 | 0.36 | 8.84 | $<2 \times 10^{-16}$ | 25.0 | [12.6–50.7] | 8.84 |
| Accuracy | 2.80 | 0.67 | 4.21 | $2.61 \times 10^{-5}$ | 16.4 | [4.4–61.1] | 4.21 |
| Precision × Accuracy | −3.14 | 0.81 | −3.89 | $9.97 \times 10^{-5}$ | 0.04 | [0.0–0.2] | 3.89 |
| Model 2: using only acceptably ambiguous market names according to CFIA (AIC = 221, McFadden's pseudeo-$R^2$ = 0.36, residual df = 235). | | | | | | | |
| Intercept | −2.11 | 0.31 | −6.90 | $5.1 \times 10^{-12}$ | | | |
| Precision | 3.39 | 0.40 | 8.55 | $<2 \times 10^{-16}$ | 29.7 | [13.5–65.0] | 8.55 |
| Accuracy | 3.21 | 0.87 | 3.68 | $2.3 \times 10^{-4}$ | 24.8 | [4.5–136.3] | 3.68 |
| Precision × Accuracy | −3.39 | 1.01 | −3.37 | $7.6 \times 10^{-4}$ | 0.0 | [0.0–0.2] | 3.37 |

## Ambiguous market names

There was no significant association between conservative estimates of mislabeling and the use of ambiguous market names in invertebrates (Fisher's exact test: $p$ = 0.4, 95% CI for Odds Ratio = 0.47–22.3) or finfish ($X^2$ = 1.9, df = 1, $p$ = 0.2). There was also no significant association between ambiguous market names and conservation status in invertebrates (Fisher's exact test: $p$-value = 1, 95% CI for Odds Ratio = 0.03-infinity). However, there was a significant relationship between conservation status and ambiguous market names in finfish, with ambiguous market names more likely to include the possibility of sale of species of conservation concern ($X^2$ = 83, df = 1, $p$ = $2.2 \times 10^{-16}$).

Logistic regression was used to analyze the relationship between precision (precise *vs* ambiguous labels) and accuracy (properly labeled *vs* mislabeled, using conservative estimates only), on conservation status (least concern *vs* conservation concern) in finfish (Table 4). The model including the interaction term had significantly lower AIC than that without. Results were similar whether including all forms of ambiguity ($n$ = 271 samples with data for all three criteria) or only those with ambiguous labels deemed acceptable by the CFIA ($n$ = 239); only the latter is discussed below (see comparisons in Table 4). The McFadden pseudo-$R^2$ for the model was reasonably high at 0.36. All three terms were significant, with precision having higher importance than accuracy, and accuracy higher importance than the interaction term. Removing the interaction term from the model provided a slightly worse fit (McFadden's pseudo-$R^2$ = 0.31); in this model only precision remained significant.

## DISCUSSION

In this study we report that (1) invertebrate and finfish products are mislabeled in Calgary, and to a degree consistent with other Canadian cities; (2) semantic mislabeling inflates estimates of mislabeling in both invertebrates and finfish, but product substitution is still a relevant concern; (3) product substitutions and legally ambiguous labels hide from

consumers finfish species of conservation concern—but ambiguous labeling is the more important predictor; and (4) contrary to expectations, legally ambiguous labeling did not appear to reduce mislabeling. Our study adds to the growing literature on Canadian seafood mislabeling (*e.g.*, *Wong & Hanner, 2008*; *Hanner et al., 2011*; *Naaum & Hanner, 2015*; *Lifescanner & SeaChoice, 2017*; *Muñoz-Colmenero et al., 2017*; *Oceana Canada, 2017*; *Hu et al., 2018*; *Levin, 2018*; *Shehata et al., 2018*, *2019*; *Thurston & Wilmot, 2019*; *Cawthorn et al., 2021*; *Morris, 2020*; *Thurston, 2021*; *Nijman & Stein, 2022*; *Rathnayake, 2022*) and is the first to compare vertebrates to invertebrates, and to examine the consequences of legally ambiguous naming practices. Most studies have focused on coastal or western Canadian cities; this is the first (beyond an initial description in *Morris (2020)* to a faith-based audience) to focus on a landlocked prairie city in Canada where one may expect higher rates of mislabeling due to less interaction with marine life, and is to date the only study with sample sizes sufficient to compare Canadian finfish and invertebrates.

Calgary's finfish mislabeling is, like in the rest of Canada, a continuing concern. Our total mislabeling of just over 32% for Calgary finfish is on the lower end of that reported on average for Canadian-wide studies (41% in *Hanner et al., 2011*; 44% in *Levin, 2018*; 46% in *Thurston, 2021*) but is higher than reported for some individual cities (*e.g.*, 17.5% to 42.9% in *Hanner et al., 2011*; 24.1% to 46.2% in *Shehata et al., 2019*), and is above that for registered processing plants or importers (*Shehata et al., 2019*). Interestingly, being far from the ocean did not produce rates appreciably higher than those reported for Canadian coastal cities: Victoria (67% of 15 samples), Halifax (38% of 89 samples), or Vancouver (25% of 285 samples, or 26% of 84 samples) (*Hu et al., 2018*; *Levin, 2018*). Although speculative, mislabeling largely affected products that have had species-specific distinguishing features removed (*e.g.*, sushi, breaded fillets) or manipulated (*e.g.*, flesh color), suggesting that coastal communities can be as easily misled as landlocked communities.

There is good reason to question total mislabeling percentages. For instance, Oceana Canada's high estimates of nearly 1 in 2 mislabeled products (*Thurston & Wilmot, 2019*; *Thurston, 2021*) includes all samples of Hamachi or yellowtail. This is because, on the Fish List, yellowtail is an acceptable name for *Limanda asper*, while Hamachi is not an acceptable name at all; yet in Asian cuisine yellowtail or Hamachi are internationally recognized labels for the Japanese amberjack (*Seriola quinqueradiata*). Any informed consumer of sushi would be surprised if their Hamachi contained a flatfish—in this case it is the Fish List that is misleading and not the market name itself. Is such semantic mislabeling really on par with the substitution of tuna for escolar, or the substitution of snapper for tilapia? Product substitution (the replacement of one species for another) has known economic, health, and conservation concerns (*Silva, Hellberg & Hanner, 2021*); the consumer believes they are eating one thing but are in fact receiving something else. While semantic mislabeling could include the sale of species of health or conservation concerns, the consumer would likely not be surprised to discover the identity of their food; health and conservation concerns are already implicit in the label. Our data adds to the growing concerns over grouping semantic mislabeling with product substitution (*Cheney, 2018*; *Hu et al., 2018*)—semantic mislabeling is relatively easy to fix by updating the Fish List, makes
a serious problem seem worse than it actually is, and can undermine confidence in mislabeling reports. Removing semantics from mislabeling caused our estimates to drop from around 1 in 3 mislabeled products to around 1 in 5– substantially below the average typically reported for Canada. Indeed, a global meta-analysis of finfish mislabeling that focused on product substitution estimated a world-wide posterior mean of 23% (95% HDI 19–28%), placing Calgary comparatively on the lower end of the global mislabeling spectrum (*Luque & Donlan, 2019*).

It is possible however that we are over- or underreporting the true nature of product substitution in Calgary, for two reasons. First, sample design was similar to that normally conducted in the literature—uncontrolled sampling of whatever caught the whims of the sampler. Some studies (*e.g.*, *Thurston, 2021*) focus only on species known to be mislabeled from other studies, whereas in our study students began with little *a priori* knowledge of what finfish are likely to be mislabeled. They sampled largely from what was cheap or readily available, particularly samples labeled tuna or salmon. More expensive amberjack or snapper were less well represented. A study design that truly represents what is available on the market would provide a better estimate of mislabeling, but is not typical in the literature. Second, it is important to think of mislabeling in the context of hypothesis testing. When a DNA barcode clearly identifies product substitution, we have compelling evidence to reject our null hypothesis that the product was labeled appropriately. But when DNA barcoding returns multiple species identifications, one of which fits the label, we fail to reject our hypothesis—but the sample could very well be from one of the other species that shares the barcode. Shared barcodes using *COI* is a known problem in many finfish, including tuna, with barcodes from other genes being developed to increase species discrimination in these systems (*Antil et al., 2023*; *Emmi, Fatusin & Hellberg, 2023*).

Little has been done on invertebrate mislabeling in Canada. Here we report, barring semantic mislabeling, an average of 20.2% product substitution. Our rates were squarely within the global average of 22% (95% HDI 15–30%) (*Luque & Donlan, 2019*). Shrimp had low levels of mislabeling and, despite the tremendous species diversity potentially marketed as shrimp according to the Fish List, it was almost entirely *Penaeus vannamei*, findings consistent with that reported by *Rathnayake (2022)* for Ontario. Crab and cephalopod products had quite high mislabeling rates. Although there is little to compare across Canada, it is interesting that *Shehata et al. (2018)* similarly reported a species of the finfish *Nemipterus* marketed as crab. Both *Nemipterus* and *Evynnis* species have been reported in Indonesia in cases of surimi mislabeling (*Abdullah et al., 2019*). An investigation into mislabeling of surimi as crab has not, to our knowledge, been done in North America.

Contrary to expectations, ambiguity in market names did not decrease mislabeling, but neither did it increase it. There were however notable within-species exceptions. Products sold as Atlantic salmon were always *Salmo salar*, whereas products sold as salmon which, according to the Fish List, must be *Salmo salar* were often other species (see for an American example *Kroetz et al., 2020*). This is an unusual example of ambiguity, in that the market name is precise (salmon refers to a single species) but in common vernacular it is ambiguous. This pattern warrants further investigation, as it could suggest a means to

combat some forms of mislabeling. On the other hand, two of 19 ambiguously-named cod were mislabeled whereas four of 12 precisely-named Atlantic/Pacific cod were mislabeled; larger sample sizes are required to determine whether removing ambiguity would help in this and other systems.

Not surprisingly, mislabeled products tended to include species of conservation concern, such as European eel being sold as unagi (*Nijman & Stein, 2022*; *Goymer et al., 2023*). Although less discussed in the literature, our data suggests ambiguous market names have a larger effect on the sale of species of conservation concern than mislabeling. Ambiguity in the label "snapper" is a global problem, covering over the sale of 67 species of fish from 16 families with an overall mislabeling rate of 40% (*Cawthorn, Baillie & Mariani, 2018*). *Hu et al. (2018)* point out that the ambiguous market name of tuna can mask cheaper, mercury-laden, or critically endangered tuna species from consumers. The relationship between ambiguous market names and conservation was not evident amongst invertebrates—largely because the conservation status of most invertebrates sold on Canadian markets is not known. However, the most common species sold under the ambiguous name of shrimp is a farmed species associated with the destruction of mangrove swamps, release of antibiotics into natural ecosystems, and the spread of disease (*Korzik et al., 2020*; *Macusi et al., 2022*)—consequences the informed consumer is prevented from appreciating when ambiguous market names are used.

## CONCLUSIONS

Approximately one in five invertebrate and finfish products sampled in Calgary were not what the labels claimed them to be. That is, even consumers attempting to make sustainable decisions in their food choices are unwittingly eating other species, and sometimes species of conservation concern. Although we know little about the conservation status of most invertebrate species sold on Calgary markets, there were notable examples of invertebrates being substituted for endangered finfish. More strikingly, finfish were more likely to be of conservation concern if they were either mislabeled or had legally ambiguous market names. The focus in the literature to date has been on product substitution, and rightly so. However, although the consumer cannot know if they are eating a substituted product, they can know whether they have purchased an ambiguously-named product; this suggests a means by which consumers can "vote with their wallet" to change market name practices. A powerful justification for the use of ambiguous market names has been taxonomic uncertainty and morphological similarity amongst taxa such as the rockfishes or prawns; but this is not true for all cases of ambiguity (*e.g.*, salmon or cod) and will become less justifiable as DNA-based identification methods become cheaper, easier, and more available (*Antil et al., 2023*).

## ACKNOWLEDGEMENTS

The authors wish to thank the 64 ECOL 529 (2014; 2016), 128 Zoology 401 (2019; 2020), 21 BIOL4310 (2017; 2019), and 132 BIO 211 (2016-2020) students who participated in seafood collection, preparation for DNA barcoding, and analysis of the results. Thanks to Arminty Clarke, Raylene Dunn, and Rachel Nottrodt for their support of this work

through co-design of educational materials and facilitation of student projects and collaboration at the University of Calgary. Heather Clitheroe provided guidance on the University of Calgary aspects of this project and logistics, and Jordann Fernandes organized and summarized initial University of Calgary student work on invertebrates.

### Funding

This work was done through two Alberta i@Home grants held by Matthew R. J. Morris, and Mindi M. Summers and Sean M. Rogers. The funders had no role in study design, data collection and analysis, decision to publish, or preparation of the manuscript.

### Grant Disclosures

The following grant information was disclosed by the authors:
Alberta i@Home.

### Competing Interests

The authors declare that they have no competing interests.

### Author Contributions

- Matthew R. J. Morris conceived and designed the experiments, performed the experiments, analyzed the data, prepared figures and/or tables, authored or reviewed drafts of the article, and approved the final draft.
- Mindi M. Summers conceived and designed the experiments, performed the experiments, prepared figures and/or tables, authored or reviewed drafts of the article, and approved the final draft.
- Morgan Kwan conceived and designed the experiments, performed the experiments, analyzed the data, prepared figures and/or tables, and approved the final draft.
- Jonathan A. Mee conceived and designed the experiments, performed the experiments, authored or reviewed drafts of the article, and approved the final draft.
- Sean M. Rogers conceived and designed the experiments, performed the experiments, authored or reviewed drafts of the article, and approved the final draft.

### DNA Deposition

The following information was supplied regarding the deposition of DNA sequences:
The 347 sequences in the Supplemental File are available at BOLD (boldsystems.org).

### Data Availability

The product information, including BOLD identifier, collector name, collection date, vendor name, vendor type, market name, DNA barcode sequence, species identity determined through DNA barcoding, type of mislabeling are available in the Supplemental File.

## Supplemental Information

Supplemental information for this article can be found online at http://dx.doi.org/10.7717/peerj.18113#supplemental-information.

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
