# Peer review of "Mislabeled and ambiguous market names in invertebrate and finfish seafood conceal species of conservation concern in Calgary, Alberta, Canada"

_PeerJ, doi:10.7717/peerj.18113_

## Round 0.1 · original submission · Minor Revisions

We have received two very positive reviews for your manuscript. Both reviewers found it to be innovative and to provide important insights on the issue of seafood mislabeling that have important implications.

They offered several minor comments that will help further improve your manuscript. For instance, reviewers suggested to provide additional details regarding the samples and their handling, as well as to be more explicit about how this study builds on previous studies about Canadian seafood mislabeling.

Reviewer 1 ·

Basic reporting

This paper presents significant findings that contribute to the issue of seafood mislabeling.
While the field has seen recent publications, this study provides a fresh perspective by focusing on legally ambiguous market names relating to Canada, which provides a new perspective. This approach makes the manuscript accessible to an audience interested in the policy implications and consumer protection issues. The Introduction effectively sets the stage by clearly defining the subject and specifying the intended audience. It compellingly motivates the study by highlighting the importance of precise market names for seafood products in conservation and consumer trust.
The approach and the analyses used in this work can be considered satisfactory. Figure are of good quality. Results are very clearly presented and the organization of the paper meets the criteria of the journal. The conclusions are drawn appropriately based on the data presented. The study is original. Language is clear and understandable. The manuscript covers significant ground in marine biology, conservation, and food safety, making it relevant to a wide range of readers from different fields.

Experimental design

The content of the manuscript aligns perfectly with the aim and scope of the journal. The survey methodology is thorough and designed to ensure comprehensive and unbiased coverage of the subject. The sampling of both invertebrate and finfish products over several years adds depth to the study. Sources are adequately cited, with appropriate use of quotes and paraphrasing. The references to boldsystems.org and the Canadian Food Inspection Agency Fish List are particularly pertinent. The manuscript is logically organized into coherent sections, each building on the previous one. This structure facilitates a clear understanding of the study's progression and findings.

Validity of the findings

The argument is well-developed and robustly supported, meeting the goals outlined in the Introduction. The statistical analysis, including logistic regression, enhances the reliability of the results. The Conclusion effectively identifies unresolved questions and future research directions, particularly the need for more precise and unambiguous naming conventions to safeguard consumers and to support sustainable fisheries.

Additional comments

The manuscript is a significant contribution to the field, addressing the critical issue of seafood mislabeling with a unique focus on market name ambiguity. The study's implications for conservation and consumer protection are well-articulated, making it a valuable resource for both researchers and policymakers.

However, in my opinion, there are specific improvements that could enhance the manuscript. Although the authors mention in the methods section that they followed the guidelines outlined by Naaum et al. (2015), it would be beneficial to include additional details about the samples. This should encompass information such as the type of sample (e.g., whole, fillet, canned, prepared meal, sashimi, smoked, dried), its processing level (e.g., head off, frozen status), and its origin (including catch method, farmed or wild designation). Moreover, the methods section should provide clarity on the procedures for sample collection, preservation, and shipment.

Reviewer 2 ·

Basic reporting

No additional comment

Experimental design

No additional comment

Validity of the findings

No additional comment

Additional comments

GENERAL COMMENTS
This manuscript reports on the first study in Canada to examine mislabeling and ambiguous market names in both invertebrate (e.g., bivalve, cephalopod, shrimp) and finfish products. As part of class assignments at three universities in Calgary, the authors compared species identity determined from DNA barcoding to the market name given in the Canadian Food Inspection Agency Fish List. They distinguished between semantic mislabeling (where the market name was not found on the Fish List, but the barcode identity was in line with what a consumer could reasonably have expected to have purchased), invalid market names (where the market name was so unusual that no legitimate inferences as to the product’s identity could be made), and product substitution (where the DNA barcode identified a species different from the market name). They found 33.9% total mislabeling occurrence and 20.2% product substitution (invertebrates) and 32.3% total mislabeling occurrence and 21.3% product substitution (finfish). Further, they determined the conservation status of the DNA-identified species (through the International Union for the Conservation of Nature Red List), and they found that product substitutions and ambiguous market names were significantly associated with the sale of species of conservation concern, with ambiguity being the more important predictor. Thus, this study is important and will have broad implications beyond the scientific community. It could help the Canadian Food Inspection Agency identify where more precise seafood names are needed in its Fish List, and it will help consumers make informed choices about which fish on the market have been sustainably harvested.

Overall, the manuscript is clearly written, and the science is sound. There are likely some biases in the samples collected (e.g., the students sampled largely from what was cheap or readily available, particularly samples labeled tuna or salmon), but the authors identify these possible biases.

My only general suggestion would be for the authors to be more explicit about how this study builds on previous studies about Canadian seafood mislabeling (e.g., Wong & Hanner, 2008; Hanner et al., 2011; Naaum & Hanner, 2015; Lifescanner & SeaChoice, 2017; Muñoz-Colmenero et al., 2017; Oceana Canada, 2017; Hu et al., 2018; Levin, 2018; Shehata et al., 2018, 2019; Thurston & Wilmot, 2019; Cawthorn et al., 2021; Morris, 2020; Thurston, 2021; Nijman & Stein, 2022; Rathnayake, 2022)—presumably because it looks at both invertebrates and finfish and looks at both mislabeling and ambiguous market names—and why Calgary (or a landlocked city/landlocked prairie city) might have been different than coastal cities (Line 350-354).

All other suggestions are minor.

SPECIFIC COMMENTS
General – I recommend stating what source you’re using for classification (e.g., FishBase for fish species names or families); there are some minor differences among authorities

Line 58 – “immanent” should be “imminent”

Line 83 – The provided link (https://inspection.canada.ca/active/scripts/fssa/fispoi/fplist/fplist.asp) didn’t work for me, although I found the CIFA Fish List here: https://active.inspection.gc.ca/scripts/fssa/fispoi/fplist/fplist.asp?lang=e. Check the link.

Line 166 – It would be useful to specify that the IUCN conservation status from the Red List (www.iucnredlist.org/) refers to global status (e.g., some species that are globally secure are still at some level of risk locally)

Line 170 – spell out BOLD on first use

Line 304 – I would capitalize Endangered to show that it is a formal status designation

Line 307-310 – Likewise for Critically Endangered and Vulnerable

Line 350-354 – What made you think that Calgary (or a landlocked city/landlocked prairie city) might be different?

Table 1 – It would be useful to briefly state in the caption what is meant by semantic mislabeling, invalid market name, and product substitution so that the table can be interpreted quickly on its own. [Table 2 and 3 captions say “See paper for definitions of mislabeling,” but non-scientists browsing through the paper because they’re interested in its implications to market labeling and conservation aren’t likely to easily find the information they need in the paper]

Table 2 – Under Crab, 1 – Threadfin Beam should be “Threadfin Bream”; under Uni, “purpuratus” should be italicized

Supplemental File – I would find this more useful if a column with DNA barcode identify was also included and, ideally, if color or other coding was added to show different labeling categories—semantic, invalid market names, and product substitution.

---

## Round 0.2 · accepted · Accept

I am pleased with the revisions made to the manuscript.